# Incorporation of Pectin into Vaterite Microparticles Prevented Effects of Adsorbed Mucin on Neutrophil Activation

**DOI:** 10.3390/ijms242115927

**Published:** 2023-11-03

**Authors:** Elena V. Mikhalchik, Liliya N. Maltseva, Roxalana K. Firova, Marina A. Murina, Irina V. Gorudko, Daria V. Grigorieva, Viktor A. Ivanov, Ekaterina A. Obraztsova, Dmitry V. Klinov, Ekaterina V. Shmeleva, Sergey A. Gusev, Oleg M. Panasenko, Alexey V. Sokolov, Nikolay P. Gorbunov, Lyubov Y. Filatova, Nadezhda G. Balabushevich

**Affiliations:** 1Department of Biophysics, Lopukhin Federal Research and Clinical Center of Physical-Chemical Medicine of Federal Medical Biological Agency, 119435 Moscow, Russia; liya.malceva.00@mail.ru (L.N.M.); rxfirova@gmail.com (R.K.F.); marina_murina@mail.ru (M.A.M.); vanov.va@inbox.ru (V.A.I.); brenka@mail.ru (E.A.O.); klinov.dmitry@mail.ru (D.V.K.); ekmos.vk@gmail.com (E.V.S.); ser_gus@mail.ru (S.A.G.); o-panas@mail.ru (O.M.P.); biochemsokolov@gmail.com (A.V.S.); niko_laygo@mail.ru (N.P.G.); nbalab2008@gmail.com (N.G.B.); 2Department of Chemistry, Lomonosov Moscow State University, 119991 Moscow, Russia; luboff.filatova@gmail.com; 3Department of Biophysics, Belarusian State University, 220030 Minsk, Belarus; irinagorudko@gmail.com (I.V.G.); dargr@tut.by (D.V.G.); 4Moscow Institute of Physics and Technology, 141700 Dolgoprudny, Russia; 5Department of Molecular Genetics, Institute of Experimental Medicine, 197376 St. Petersburg, Russia

**Keywords:** vaterite, pectin, mucin, hybrid vaterite microparticles, neutrophils, chemiluminescence, antioxidants, cytotoxicity

## Abstract

The application of vaterite microparticles for mucosal delivery depends on their interaction with mucin and immune cells. As we have shown previously, the binding of mucin onto particles enhances the generation of reactive oxygen species by neutrophils. The attenuation of the pro-oxidant effect of the bound mucin through the modification of vaterite could improve its biocompatibility. Hybrid microparticles composed of vaterite and pectin (CCP) were prepared using co-precipitation. In comparison with vaterite (CC), they had a smaller diameter and pores, a greater surface area, and a negative zeta-potential. We aimed to study the cytotoxicity and mucin-dependent neutrophil-activating effect of CCP microparticles. The incorporated pectin did not influence the neutrophil damage according to a lactate dehydrogenase test. The difference in the CC- and CCP-elicited luminol or lucigenin chemiluminescence of neutrophils was insignificant, with no direct pro- or antioxidant effects from the incorporated pectin. Unlike soluble pectin, the CCP particles were ineffective at scavenging radicals in an ABAP–luminol test. The fluorescence of SYTOX Green demonstrated a CCP-stimulated formation of neutrophil extracellular traps (NETs). The pre-treatment of CC and CCP with mucin resulted in a 2.5-times-higher CL response of neutrophils to the CC-mucin than to the CCP-mucin. Thus, the incorporation of pectin into vaterite microspheres enabled an antioxidant effect to be reached when the neutrophils were activated by mucin-treated microparticles, presumably via exposed ligands.

## 1. Introduction

Vaterite is a polymorph of calcium carbonate, with particle dimensions ranging from nanometres to tens of micrometres, a spherical surface, a highly developed internal structure, and a high porosity; it is biocompatible and biodegradable but thermodynamically unstable [1].

Vaterite has prominent mucoadhesive properties [2] suitable for mucosal delivery, including in the gastrointestinal tract. However, as was shown in our previous studies, vaterite microparticles with adsorbed mucin (CC-M), the major glycoprotein constituent of mucus, exhibit a significantly enhanced generation of reactive oxygen and halogen species (ROS/RHS) by neutrophils [3]. This is why it is of special interest to find an approach to attenuate this effect, especially in the context of chronic inflammatory bowel diseases. The prospective approaches include the incorporation of various biopolymers into vaterite microparticles, which would thereafter become more stable [4,5].

In the present study, vaterite was co-precipitated with apple pectin, resulting in hybrid microparticles. Commercially available pectin from apple pulp is a partially methyl-esterified and non-sulphated polysaccharide. Its main chain is composed of 1,4-linked α-D-galactosyluronic acid and α-1,2-L-rhamnose, and its side chains include various uncharged monosaccharides such as arabinose, galactose, and others [6]. Pectin is a major component of plant cell walls, and like many other natural polysaccharides, it varies in its molecular mass depending on the type of raw material and the method of isolation used. Beyond applications in the food industry, pectin has been studied in view of its biomedical significance: its immunoregulatory activity, anti-inflammatory effect, hypoglycaemic effect, and antibacterial activity [7]. The antioxidant activity of pectin has been evaluated with a number of methods that employ the synthetic stable free radicals ABTS^•+^ and DPPH^•^ or hydroxyl radicals, depending on pectin’s structure, and the activity was about 200-times lower in comparison with ferulic acid or Trolox [8].

Pectin has been studied as a component of delivery systems and is incorporated into gels, films, and polyelectrolyte complexes with alginate, chitosan, casein, gelatine, carboxymethylcellulose [6], inorganic antimicrobe hydroxyapatite nanoparticles for regenerative medicine [9], anti-inflammatory silica gels for gastrointestinal delivery [10], or calcium carbonate microparticles for the removal of Cu(II) and Ni(II) cations [11].

In order to study some aspects of the biocompatibility of hybrid vaterite–pectin microparticles, we evaluated their antioxidant properties, cytotoxicity, sorption of mucin, and activation of the neutrophil respiratory burst. The antioxidant effects of pectin and CCP were evaluated in a model system of ABAP–luminol, as well as at the activation of neutrophils in vitro, in comparison with CC. Mucin was adsorbed onto the particles or added into the measurement media. Since the neutrophil oxidative burst assayed with chemiluminescence could be affected by cell viability, we also assessed the cytotoxicity of the particles using the extracellular lactate dehydrogenase (LDH) activity, the formation of NETs using SYTOX Green fluorescence, and the functional activity as the amplitude of the chemiluminescent response to the second activator, phorbol 12-myristate 13-acetate (PMA).

## 2. Results

### 2.1. Antioxidant Activity of Pectin

The antioxidant activity of pectin was evaluated using the thermally decomposing compound 2,2′-azobis(2-amidinopropane) dihydrochloride (ABAP), giving peroxyl radicals ROO^•^ in the presence of oxygen. Peroxyl radicals were detected using luminol-enhanced chemiluminescence (Lum-CL) [12]. Antioxidants reacting with the peroxyl radicals would quench the chemiluminescence for some period of time, called the “induction time” or “lag-period”, which corresponds to the antioxidant capacity of the tested compound.

With the growth of the pectin concentration, the lag-period increased (Figure 1a,b) as a result of peroxyl radical trapping. The effect of sodium ascorbate as a standard antioxidant was also evaluated (Figure 1c,d), and by weight, it was 1000-times more effective than pectin.

Additionally, we analysed the interaction of pectin with hypochlorite, one of the reactive halogen species generated by neutrophil myeloperoxidase, by assessing the residual concentration of OCl^−^ ions using an iodometric titration (Figure 2). At an initial NaOCl concentration of 90 μM, the C_50_ of pectin was approximately 0.6 μg/mL.

Next, we analysed the antioxidant effects of pectin towards ROS/RHS generated by neutrophils in vitro. For this purpose, we registered the intensity of the spontaneous and PMA-stimulated luminol (Lum)- or lucigenin (Luc)-CL of neutrophils (Figure 3a,b).

The inhibition of spontaneous Lum-CL by pectin was stronger than the inhibition of Luc-CL, while the PMA-stimulated responses decreased proportionally. These effects of 1 mg/mL of pectin were not caused by neutrophil damage, according to the LDH test, and could be attributed to the scavenging of superoxide radicals (registered using Luc-CL) and hypochlorite (registered using Lum-CL) by pectin.

### 2.2. Microparticle Characteristics

The co-precipitation of pectin for the fabrication of CCP microparticles resulted in polysaccharide incorporation up to 5 % *w*/*w*. In comparison with CC, the percentage of calcite in the CCP microparticles was the same, but the diameter of the CCP microspheres was 1.8-times smaller (Figure 4), the pore size was 4.8-times less, and the total surface was 4-times greater (Table 1).

These results are consistent with the data on the properties of spherical hybrid microparticles of vaterite with other natural polyanions (heparin [13], cellulose [5], and mucin [14]).

The zeta-potential of the polyanion pectin in water was −20 ± 2 mV, and for CCP, it was also negative (−12 ± 2 mV), in contrast to CC. This indicated that the incorporated pectin was partially exposed on the surface of the CCP microparticles.

In order to evaluate the antioxidant activity of the incorporated pectin, we also used an ABAP–luminol system (like in the experiments with pectin), but no effects of CCP (or CC) microparticles in concentrations up to 1 mg/mL were registered. Indeed, this concentration corresponded to no more than 50 μg/mL of pectin, which was below the sensitivity of the method according to Figure 1.

The incorporated pectin reduced the adsorption of mucin by the CCP microparticles by 1.7-fold in comparison with CC (Table 1). Mucin from a porcine stomach with a molecular mass of ~600 kDa and a pI of 3–4 was characterised by a zeta-potential equal to −15 ± 2 mV and consisted of two fractions with hydrodynamic diameters of 30–50 nm and 200–300 nm. The negative charge of CCP microparticles could complicate its adsorption. As shown by fluorescent microscopy, the FITC-labelled mucin efficiently covered both the CC (Figure 5a) and CCP (Figure 5b) microparticles, so the difference in mucin adsorption was not crucial. Moreover, as shown in our previous study [2], mucin can partially penetrate the pores in CC microparticles.

### 2.3. Activation of Neutrophils with Microparticles

Microparticles of 2–4 μm in diameter are microbe-sized objects whose size facilitates their interaction with neutrophils, depending on the zeta-potential and the exposed ligands for neutrophil receptors. The generation of ROS/RHS was detected using Luc-CL sensitive to superoxide anions and Lum-CL in order to detect mainly hypochlorite. With regards to the potential effects of mucin adsorption, the CL was also measured in the presence of 0.5 mg/mL of mucin, and/or neutrophils were stimulated with CC and CCP microparticles pre-treated with mucin (CC-M and CCP-M).

Figure 6a illustrates the typical time-course of Lum-CL when neutrophils were stimulated with the microparticles, followed by stimulation with PMA. The integral CL intensity was presented as a percentage of the control values (Figure 6b,c). A corresponding aliquot of 0.15 M NaCl was added instead of the particles in the control. In parallel measurements, instead of a PMA addition, the cells with particles after the CL analysis were precipitated and the supernatant was collected for the LDH test. Thus, the functional activity of neutrophils after their interaction with the microparticles could be characterised by the amplitude of the CL response to PMA (Figure 6d,e), and their damage by the activity of the extracellular LDH (Figure 6f).

There were no significant differences in the intensity of the neutrophil Lum-CL responses to the CC and CCP microparticles, even when mucin was added to the medium (Figure 6b), and the same result was observed when the Luc-CL was measured (Figure 6c). One could conclude that the incorporated pectin was inefficient at the scavenging of superoxide or hypochlorite, probably because of its low concentration and/or availability. Moreover, no activation of neutrophils or cell damage caused by the incorporated pectin was registered.

The pre-treatment of CC microparticles with mucin enhanced both the Lum-CL (Figure 6b) and Luc-CL (Figure 6c), which was consistent with our previous results [3]. Nevertheless, the adsorption of mucin by CCP-M did not increase the ROS or RHS production, while CC-M significantly exceeded CCP-M in the intensity of the stimulated CL (*p* ˂ 0.05).

A comparison of the amplitude values of the PMA-stimulated CL after the activation of neutrophils with microparticles showed that only CCP-M caused a decrease in the Luc-CL value by 15% (Figure 6e), with no effect on the PMA-stimulated Lum-CL (Figure 6d) or cell damage (Figure 6f).

One could hypothesise that the pectin, which was exposed on the surface of the CCP microparticles, affected the adsorption of mucin both quantitively and qualitatively. Therefore, since pectin could be oxidised with OCl^−^ (Figure 2), we further studied the effects of the hypochlorite-modified microparticles CC-OCl and CCP-OCl on the neutrophil Lum-CL in the presence of 0.5 mg/mL of mucin, as well as without mucin (Figure 7).

The pre-treatment of the CC and CCP microparticles with OCl^−^ did not affect the neutrophil Lum-CL in the absence of mucin, while in the presence of 0.5 mg/mL of mucin, the response to CCP-OCl became significantly higher than that to CCP (*p* ˂ 0.05).

It is known that ROS and/or RHS can stimulate activated cells to form NETs and release DNA decorated with histones, elastase, myeloperoxidase (MPO), and other proteins [15]. Since pectin has shown antioxidant effects towards the ROS/RHS generated by neutrophils, it was important to study the role of free and incorporated pectin in NET formation.

NET formation was measured using the SYTOX Green staining of extracellular DNA (Figure 8a) and the Cy5-labelled antibody staining of MPO (Figure 8b) using flow cytometry. The extracellular DNA in the supernatants after the CL analysis was assessed using the fluorescence of SYTOX Green (Figure 8c).

Pectin (1 mg/mL in solution) significantly decreased the percentage of SG-positive neutrophils under the stimulation of neutrophils with PMA, a potent inducer of NETosis (Figure 8a), as well as the fluorescence of Cy5-conjugated MPO-antibody-labelled neutrophils (Figure 8b), presumably via the scavenging of ROS-/RHS-inducing NET formation. The CC microparticles did not cause NET formation and had no effect on PMA-induced NETosis. The CCP microparticles stimulated NETosis (Figure 8a,b) but did not affect PMA-induced NET formation. These data are consistent with the results obtained after the analysis of the fluorescent signal due to extracellular DNA binding to SYTOX Green in the neutrophil supernatants after the CL analysis (Figure 8c).

Moreover, unlike PMA, CC- and CCP-induced NET formation was not inhibited by soluble pectin (1 mg/mL), even after mucin adsorption: in the presence of pectin, F504/523 increased to 133 ± 23% for CCP, but did not change for CCP-M (102 ± 16%). For CC and CC-M, the effects were more significant: 250 ± 79% and 219 ± 35%, respectively.

## 3. Discussion

The development of new vehicles for mucosal delivery requires the synthesis of multifunctional particles composed of components that not only play structural roles, but also have other valuable properties. Our study was aimed at evaluating the biocompatibility of pectin–vaterite hybrid microparticles as a mucosal drug-delivery system. They can be used for intranasal [16] and peroral [17] delivery, in ophthalmology [18] and pulmonology [19].

Among the potential polysaccharides for incorporation into inorganic particles, pectin has attracted growing attention from researchers. The antioxidant properties of pectin are usually evaluated as radical-scavenging activity towards DPPH^•^, ABTS^•+^, and ^•^OH, registered in the range of 0.7–10 mg/mL [8,20].

Herein, the antioxidant activity of commercial methyl-esterified apple pectin was studied using an ABAP–luminol system sensitive to peroxyl-trapping compounds [12]. Commercial apple pectin showed activity 1000 times less than ascorbate in a comparable weight ratio. Pectin can also react with hypochlorite (90 μM) with a C_50_ of 0.6 mg/mL (Figure 2), being oxidised with the formation of carboxyl groups [21], which is important for its biological and anti-inflammatory activity [22,23] and mucoadhesivity [24].

In our experiments, 1 mg/mL of pectin in the measurement medium significantly decreased the CL of neutrophils stimulated in vitro with microparticles and PMA. The scavenging effect of pectin depends on the profile of the generated ROS/RHS and the ratio of their intra-/extracellular production. Table 2 presents the effects of pectin as a percentage of the control values (without pectin).

The intensity of the Lum-CL and Luc-CL neutrophil responses to the microparticles showed a close sensitivity to pectin, and there was a difference between the microparticles: the CCP-M-stimulated CL was inhibited more than the CC-M-stimulated response, while there was only a slight difference between CC and CCP.

In fact, these data demonstrate that the difference in the CCP-M- and CC-M-microparticle-induced activation of the neutrophil respiratory burst cannot be attributed only to the quantity of adsorbed mucin: mucin made the CL response to CC less sensitive to pectin, while the response to CCP-M was inhibited more than that to CCP. Since the percentage of Lum-CL and Luc-CL for each type of microparticle was close, it is probable that pectin can also scavenge superoxide radicals generated extracellularly, as demonstrated by other researchers [25].

There were no direct antioxidant effects of pectin incorporated into vaterite microparticles, detected neither in the ABAP–luminol model system nor in experiments with neutrophils. According to the free pectin activity detected in the range of 0.4–1 mg/mL, the effects of 1 mg/mL of pectin in CCP microparticles were below the sensitivity of the applied methods.

Nevertheless, the incorporation of pectin into CCP was able to attenuate the activation of the neutrophil oxidative burst stimulated by mucin-pre-treated microparticles, in comparison with CC microparticles.

The most probable mechanism underlying this effect of incorporated pectin might be mediation by neutrophil pattern-recognition receptors (PRRs) [26,27]. Pectin participated in mucin adsorption, both in a quantitative and qualitative manner, presumably via its carboxyl groups, since the oxidative modification of CCP with OCl^−^ resulted in an increase in the neutrophil-activating activity of CCP-OCl in the presence of 0.5 mg/mL of mucin in the measurement medium. The partial oxidation of pectin on the surface of CCP could increase the number of carboxyl groups, which would influence its mucoadhesivity via inter-molecular interactions and hamper the penetration of negatively-charged mucin into the pectin layer [24]. At the same time, pectin could prevent the formation of Ca(OCl)_2_ on the particles’ surfaces, followed by a reduced activation of neutrophils detected in the samples with CC-OCl (Figure 7).

No cytotoxic or cell-inactivating effects of the incorporated pectin were found under our experimental conditions. However, there was a difference between the CC and CCP microparticles in the induction of NETosis. Free pectin significantly decreased the formation of NETs stimulated with PMA, presumably via its antioxidant activity. However, CCP microparticles were more capable as inducers of NETosis than CC, and since the neutrophils remained alive and functionally active, this could be characterised as a vital NETosis [28]. Moreover, unlike PMA, CC- and CCP-induced NET formation was not inhibited by soluble pectin, even after mucin adsorption. We suggest that pectin adsorption onto the particles, not pectin’s antioxidant properties, underlies these effects. Indeed, this is more proof of a non-lytic mechanism of NET formation, caused by exposed pectin on the particles, and this mechanism is known to be independent of ROS formation [29].

It is difficult to predict the in vivo benefits of vaterite microparticles modified with pectin (CCP) on the basis of in vitro experiments, but there are some points that look encouraging. Thus, with regard to the development of new drug-delivery systems, the hybrid pectin–vaterite microparticles demonstrated a number of valuable properties. Vaterite and pectin are natural materials, with both being biocompatible and biodegradable [30,31,32,33]. Moreover, they are pH-sensitive due to the vaterite matrix [34]; thus, the pH-sensitivity of the drug-delivery system could play an important role in the colon of patients with inflammatory bowel disease (IBD), where the pH is lower than in a normal colon (up to 2.3 vs. 6.8) [35].

We have shown that, in line with vaterite microparticles [2], hybrid microparticles with pectin easily bind to mucin (Table 1 and Figure 5), which could provide an affinity to mucosa. Compared with vaterite microparticles, CCP microparticles are negatively charged, which could increase their adherence to positively-charged proteins in inflamed tissue [36]. In addition, the diameter of the CCP microparticles was less than the diameter of the vaterite (CC) microparticles (Table 1), which could facilitate their local effects in mucosa [37].

In the absence of mucin in the measurement buffer, there was no difference between CC and CCP in their activating ability towards neutrophils (Figure 6b). With mucin, CC activated neutrophils to a higher extent than CCP. We consider this property of CCP very promising, since it could allow for the prevention of excessive ROS production by neutrophils in inflamed colon tissue [38], for example. At the same time, the CC and CCP particles did not cause neutrophil disfunction, even after mucin adsorption (Figure 6d,e). An increase in the percentage of SG-positive neutrophils (Figure 8a,c), in line with the data from the LDH test (Figure 6f) and on a neutrophil PMA-induced response (Figure 6d,e), could be attributed to vital NETosis, that is, to early nonlytic NET formation in response to pathogen-associated molecular patterns, without neutrophil death. The role of NETs in inflammation, for example, in IBD, can be beneficial or negative [39], and further research in vivo is needed. 

## 4. Materials and Methods

### 4.1. Reagents Used

CaCl_2_ (≥93.0%), Na_2_CO_3_ (≥99.0%), CaCl_2_, NaCl, EDTA, Tris, Histopaque (1.077, 1.119), DMSO, NaOCl, FITC, phorbol 12-myristate 13-acetate (PMA), 2,2′-azobis(2-amidinopropane)dihydrochloride (ABAP), luminol, lucigenin, mucin from a porcine stomach (type III, with 0.5–1.5% bound sialic acid), pectin from apples (methyl-esterified), sodium L-ascorbate, and all salts and solvents for the preparation of solutions were purchased from Sigma-Aldrich (St. Louis, MO, USA). The Krebs–Ringer buffer solution was purchased from Merck (Kenilworth, NJ, USA). The Olvex-LDH kit was purchased from LLC Olveks Diagnosticum, SPb, Russia. The SYTOX Green stain was purchased from ThermoFisher Scientific (Waltham, MA, USA).

The anti-MPO antibodies conjugated with Cy5 were kindly provided by A.V. Sokolov and N.P. Gorbunov. The fluorescein isothiocyanate-labelled mucin (FITC-mucin) (17 ± 2% modified amino groups) was kindly provided by N.G. Balabushevich.

### 4.2. Fabrication of Microparticles

Vaterite microparticles (CC) were synthesized by mixing solutions of CaCl_2_ and Na_2_CO_3_ in 0.05 M Tris buffer (pH 7.0) as described earlier [4]. For the synthesis of vaterite–pectin (CPP) microparticles, an apple pectin solution was added up to 5 mg/mL. Finally, the particles were washed with double-distilled water and lyophilized.

### 4.3. Evaluation of Physical–Chemical Properties of Microparticles

For scanning electron microscopy, the microparticles were deposited onto silicon wafers and characterised using a Zeiss Merlin microscope (Zeiss, München, Germany). The SEM accelerating volage was 1–2 kV. The diameter of the microspheres (d_particles_) was calculated using the Gwiddion software version 2.0.

The porosity of the microparticles was studied using low-temperature nitrogen adsorption–desorption on an ASAP-2000 setup (Micromeritics, Norcross, GA, USA). The specific surface area was estimated using the BET method, and the pore size was calculated using the Barrett–Joyner–Halenda (BJH) method.

The ζ-potential of the biopolymers and microparticles and the hydrodynamic diameter of mucin was measured using a Zetasizer Nano ZS analyser (Malvern Instruments, Malvern, UK) with He-Ne laser, 5 mW, 633 nm, and estimated using the Smoluchowski equation. The Universal “Dip” cell (ZEN1002) was applied to measure the ζ-potential.

An X-ray powder diffraction analysis was performed using a Miniflex 600 (Rigaku, Japan) with a silicon drift detector on powdered samples using Cu Ka 1.5418 Å radiation and a Ni filter with 2ɵ from 20° to 60°.

### 4.4. Adsorption of Mucin

For the assay of the adsorption of mucin, a mixture of 40 mg/mL of microparticles and 1 mg/mL of mucin from a porcine stomach in 0.05 M Tris buffer (pH 7.4) was incubated for 30 min at 25 °C. Then, the particles were precipitated using centrifugation (5 min at 5000 rpm). The residual concentration of mucin in the supernatants was assayed using optical absorbance at 214 nm or 260 nm. In the experiments with CCP microparticles, mucin was assayed with gel filtration using a BioFox 17 SEC column (Bio-Works, Uppsala, Sweden) as described earlier [2]. The pre-treatment of CC and CCP microparticles with commercial or FITC-labelled [40] mucin was carried out as follows: equal volumes of suspended microparticles (20 mg/mL) were mixed with 10 mg/mL of a mucin solution, incubated at 37 °C, and stirred for 1 h, and then the particles were precipitated and washed with 0.15 M NaCl.

### 4.5. Fluorescent Microscopy

Fluorescent images of the microparticles treated with FITC-labelled mucin (M-FITC) were obtained using a Nicon Ni-E microscope (Japan, Tokyo), with a filter FITC (excitation from 465 to 495 nm; emission from 515 to 555 nm).

### 4.6. Antioxidant Activity in ABAP–Luminol System

The antioxidant activity of free pectin and the microparticles was assayed using the Lum-CL, measured using the luminometer Lum 1200 (DiSoft, Moscow, Russia).

A total of 1 mL of measurement media, namely 50 μM luminol in 50 mM Tris-HCl buffer with 0.15 M NaCl and 1 mM EDTA (pH 8.5), was incubated in a luminometer cell for 5 min at 37 °C. Then, ABAP was added up to 1 mM. When chemiluminescence developed, pectin up to 2 mg/mL or microparticles up to 1 mg/mL were added, the solution was mixed quickly, and the CL was registered in continuous mode. The antioxidant effect was evaluated by the duration of the lag-period (Lag) after sample injection. Sodium ascorbate was used as a standard antioxidant.

### 4.7. Neutrophil Isolation

Human neutrophils were isolated from the normal blood of 6 volunteers aged 35–60 years without inflammatory diseases, on the basis of their informed consent and agreement and approved by the local ethics committee. The blood was layered over a double gradient of Histopaque (1.077/1.119 g/L), and after centrifugation for 45 min, the neutrophils were collected, washed, and resuspended in Krebs–Ringer solution. The cell concentration was assayed by direct counting using a Goryaev chamber, with a neutrophil purity of >95%.

### 4.8. Chemiluminescent Assay of Neutrophil Activation

The CL was measured in 0.5 mL of Krebs–Ringer solution (pH 7.4) with 0.2 mM luminol (Lum-CL) or lucigenin (Luc-CL) and neutrophils (0.25–0.35 × 10^6^ cells/mL). Microparticles and pectin were added in a concentration of 1 mg/mL if not otherwise indicated. If necessary, the measurement medium was supplemented with 0.5 mg/mL of mucin. The intensity of the CL response was assayed as an integral value for 20–40 min and presented as a percentage of the control values (without particles or pectin). When the CL maximum was reached, some cuvettes were centrifugated at 2000 rpm for 5 min, and the supernatants were collected for a further assay of the LDH activity and SYTOX Green fluorescence. PMA (0.3 μM) was added to the other cuvettes at the same moment and the amplitude of the CL was measured.

### 4.9. Treatment with NaOCl

Pectin solutions of 0.02, 0.2, or 2.0 mg/mL in 0.15 M NaCl were mixed with an equal volume of 180 μM NaOCl and incubated for 30 min at 27 °C. Then, the concentration of HOCl was evaluated using an iodometric titration. The results are presented as a percentage of the control (without pectin).

Microparticles (10 mg/mL) were mixed in an equal volume with 100 μM NaOCl (pH 7.0) and incubated for 30 min at 37 °C with mild shaking. Then, the particles were precipitated at 5000 rpm for 5 min, washed twice, and resuspended in 0.15 M NaCl.

### 4.10. Lactate Dehydrogenase Assay

The cytotoxicity assay was based on an evaluation of the activity of extracellular LDH, by measuring the rate of NADH utilisation using an Olvex-LDH kit.

### 4.11. Assay of Extracellular DNA by SYTOX Green Fluorescence

A total of 5 μL of 50 μM SYTOX Green in DMSO was added to 500 μL of supernatant obtained after the CL analysis (p. 4.6) and incubated for 15 min at 25 °C. Then, an equal volume of Krebs–Ringer solution was added and the F504/F523 fluorescence was measured using a solar fluorometer (Minsk, Belarus).

### 4.12. Detection of Neutrophil Extracellular Traps by Flow Cytometry

The formation of neutrophil extracellular traps (NETosis) was detected using flow cytometry as described elsewhere [41]. Monoclonal anti-MPO antibodies were obtained after the immunization of mice [42] and labelled with the NHS-ester of Cy5 (Lumiprobe, Moscow, Russia) according to the recommendations of the manufacturer. The neutrophils were incubated with pectin and CC or CCP microparticles in a concentration of 1 mg/mL for 30 min at 37 °C, and then, if necessary, PMA was added (25 nM, the standard NET formation inducer) and the solution was incubated for another 30 min at 37 °C. Then, the DNA-binding dye SYTOX Green (50 nM), which is impermeant to live cells, and anti-MPO antibodies (14.5 μg/mL) conjugated with the Cy5 fluorescent label were added to the samples, which were incubated for 5 min at room temperature in the dark. To excite the SYTOX Green fluorescence, a laser with a wavelength of 488 nm was used; for registration, a filter of 520 ± 20 nm (FITC channel) was used. Cy5 excitation was carried out using a laser with a wavelength of 638 nm; for registration, a filter of 660 ± 20 nm (APC channel) was used. The resulting cell suspension was analysed using a CytoFocus flow cytometer (Healicom, Guangdong, China). At least 30,000 neutrophils were recorded in each sample, and the data obtained were analysed using the software supplied with the flow cytometer. The following quantitative characteristics of NETosis were chosen: the percentage of SYTOX Green-positive cells in a selected neutrophil population based on forward and side-scatter characteristics, and the average fluorescence intensity of neutrophils carrying Cy5-conjugated anti-MPO antibodies on their surface.

### 4.13. Statistics

The statistical significance levels were calculated with Student’s *t*-test using STATISTICA software version 12 (StatSoft, Inc., Tulsa, OK, USA). *p*-values ˂ 0.05 were assumed to be significant. The data are represented as the mean value ± standard deviation.

## Figures and Tables

**Figure 1 ijms-24-15927-f001:**
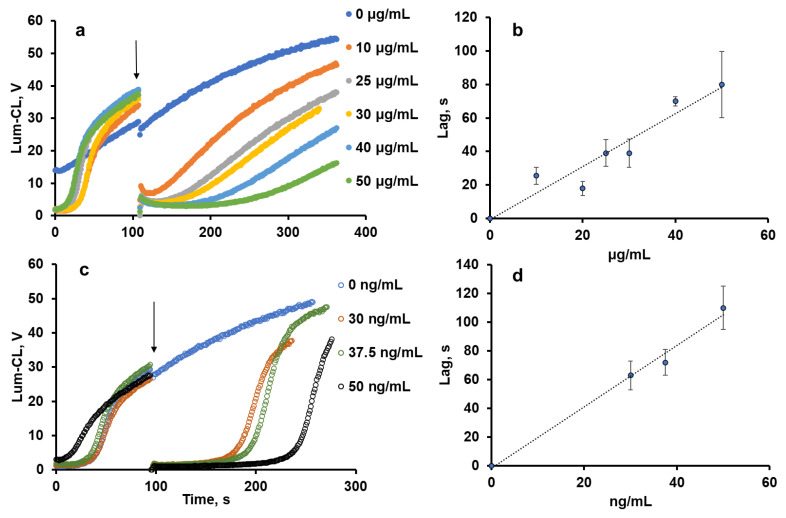
Effects of pectin (**a**,**b**) and ascorbate (**c**,**d**) concentrations on the profile of Lum-CL intensity (**a**,**c**) and the duration of the lag-period (**b**,**d**) in the model system (50 μM luminol in 50 mM Tris-HCl buffer (pH 8.5) with 0.15 M NaCl and 1 mM ABAP). Arrows indicate the time-point of the pectin or ascorbate addition.

**Figure 2 ijms-24-15927-f002:**
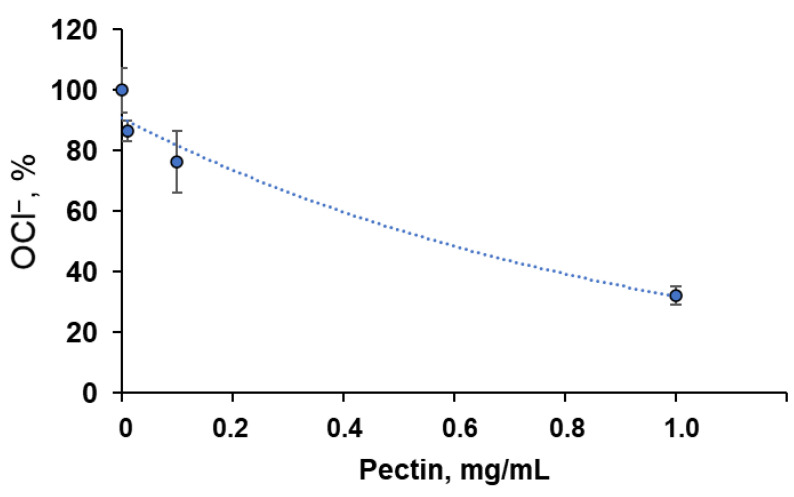
Dependence of residual OCl^−^ concentration on pectin concentration. An amount of 90 μM NaOCl was incubated with various concentrations of pectin for 30 min at 27 °C. The results are presented as a percentage of the control values (without pectin).

**Figure 3 ijms-24-15927-f003:**
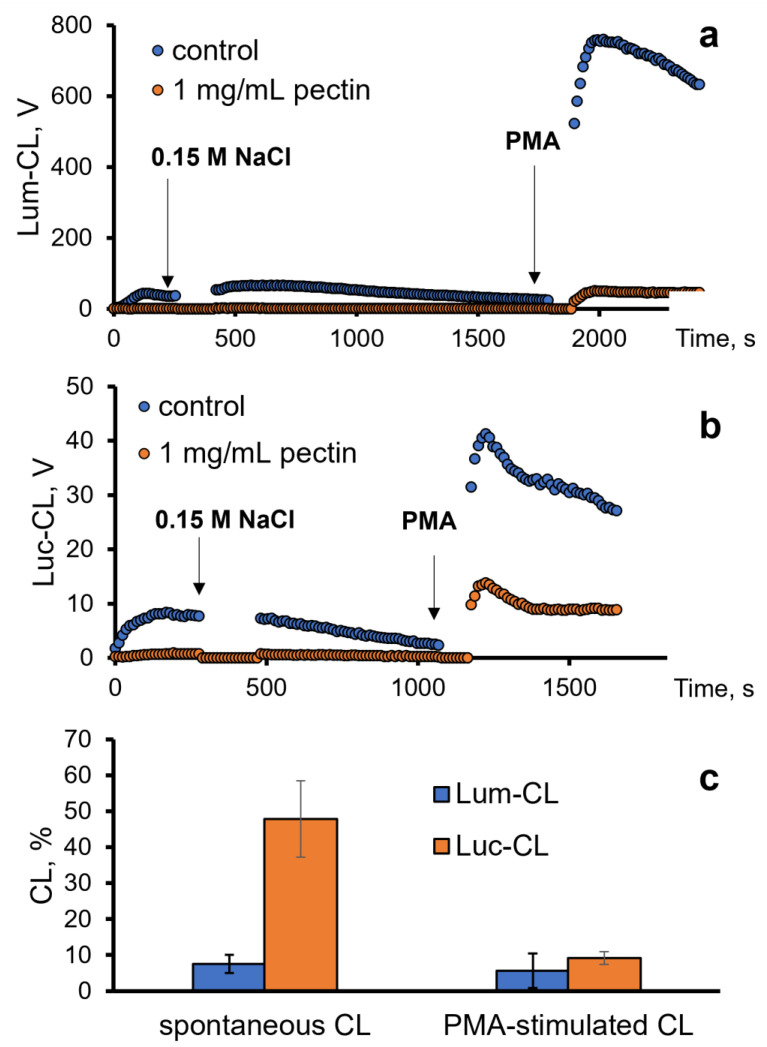
Profile of the Lum-CL (**a**) and Luc-CL (**b**) response of neutrophils: spontaneous (after addition of 0.15 M NaCl) and PMA-stimulated, both in the presence of 1 mg/mL of pectin or without pectin (control). (**c**) Intensity of spontaneous and PMA-stimulated Lum-CL and Luc-CL as a percentage of the control values.

**Figure 4 ijms-24-15927-f004:**
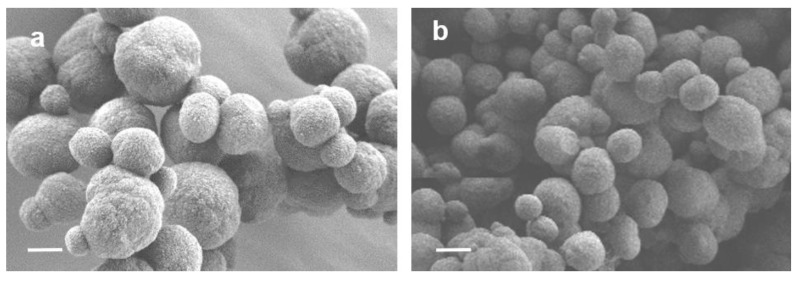
Scanning electron microscopy images of CC (**a**) and CCP (**b**) microparticles. Scale bar: 2 μm.

**Figure 5 ijms-24-15927-f005:**
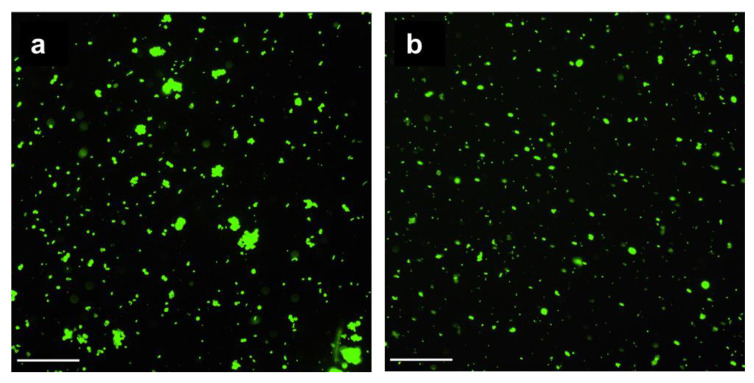
Fluorescent images of CC and CCP microparticles treated with mucin-FITC: CC-M-FITC (**a**) and CCP-M-FITC (**b**). Scale bars: 100 μm.

**Figure 6 ijms-24-15927-f006:**
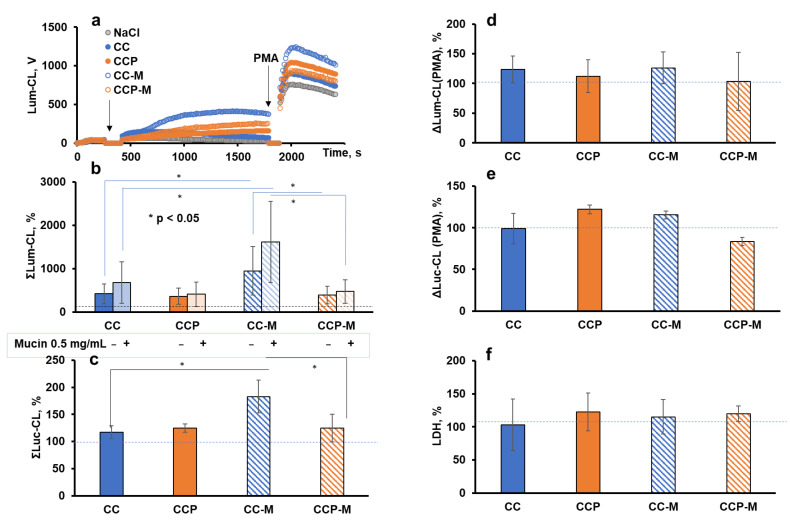
Activation of neutrophils from healthy donors (*n* = 6) with CC, CCP, CC-M, or CCP-M microparticles (1 mg/mL) in Krebs–Ringer solution: typical time-course of Lum-CL stimulated by microparticles followed by PMA (**a**); integral CL intensity of neutrophil response to microparticles enhanced by luminol (**b**) or lucigenin (**c**), presented as a percentage of the control values; the amplitude of Lum-CL (**d**) and Luc-CL (**e**) neutrophil response to PMA as a second stimulus; and (**f**) the activity of extracellular LDH after neutrophil response to microparticles. Mucin in the amount of 0.5 mg/mL was added into the medium in a number of measurements (**b**). * *p* ˂ 0.05 according to a *t*-test. The dashed lines correspond to the control value (without particles).

**Figure 7 ijms-24-15927-f007:**
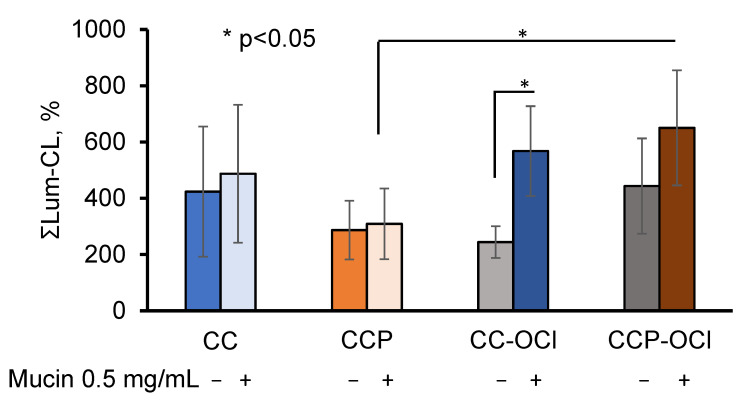
Integral intensity of Lum-CL neutrophil response to CC, CCP, CC-OCl, and CCP-OCl microparticles in the medium, with and without 0.5 mg/mL of mucin. The results are presented as a percentage of the control (without pectin).

**Figure 8 ijms-24-15927-f008:**
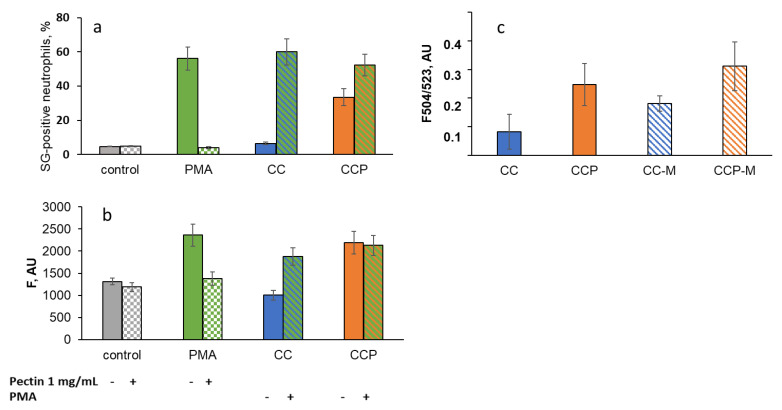
Effects of CC and CCP microparticles and 1 mg/mL of pectin on NETosis as a percentage of SYTOX Green-positive (SG-positive) neutrophils (**a**) and fluorescence of Cy5-conjugated MPO-antibody-labelled neutrophils (**b**); PMA-stimulated neutrophils were analysed as a positive control (flow cytometry data). (**c**) The fluorescence of SYTOX Green that became bound to intracellular DNA was detected in the supernatants of neutrophils stimulated with CC, CCP, CC-M, or CCP-M microparticles after the CL analysis.

**Table 1 ijms-24-15927-t001:** Physical–chemical characteristics of microparticles.

Particles	Content, %		D_particle_, μm	S, m^2^/g	D_pore_, nm	ζ-Potential, mV	Adsorption of Mucin, mg/g
	Vaterite to CaCO_3_	Pectin by Weight					
**CC**	97.5	-	3.9 ± 0.6	22 ± 3	18.4	2 ± 1	7 ± 2
**CCP**	98.0	2–5	2.1 ± 0.5	95 ± 10	3.8	−12 ± 2	4 ± 1

**Table 2 ijms-24-15927-t002:** Effect of pectin (1 mg/mL) on the CL response of neutrophils to microparticles, as a % of the control value (without pectin).

Samples	Sample-Stimulated CL, %	
	Luminol	Lucigenin
CC	21.6 ± 1.6	15.1 ± 1.6
CCP	17.0 ± 3.9	20.9 ± 2.9
CC-M	52.4 ± 10.9	41.9 ± 10.8
CCP-M	6.0 ± 3.4	11.9 ± 6.6
0.15 M NaCl (control)	2.4 ± 0.7	8.1 ± 4.3

## Data Availability

The data are available from the authors.

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
