# Peer review of "Incorporation of Pectin into Vaterite Microparticles Prevented Effects of Adsorbed Mucin on Neutrophil Activation"

_ijms, 2023, doi:10.3390/ijms242115927_

Round 1

Reviewer 1 Report

Authors described that the incorporation of pectin into vaterite microspheres enabled to reach antioxidant effect when neutrophils were activated by mucin-treated microparticles, presumably via exposed ligands. Reviewer thinks that the data demonstrating the in vivo benefit of pectin into vaterite microspheres is weak. 

In comparing CCP and CC, there was no difference in cytotoxicity, and while neutrophil activity decreased with CCP, SG-positive neutrophils, an indicator of Netosis, increased with CCP. There is a lack of consideration regarding the relationship between Netosis and neutrophil activity. CCP appears to be toxic in vivo, making it difficult to find consistency in the results.

It was understood that free pectin has an antioxidant effect, and it was also understood that Table 2 shows that free pectin and CCP-M have a synergistic effect on neutrophil activity. Therefore, regarding Netosis, the synergistic effect of CCP-M and free pectin should be verified as well.

The + and - symbols in Figure 8 are misaligned.

Author Response

Response to Reviewer 1.

We are grateful to the Reviewer for constructive comments and valuable suggestions, which help improve the quality of the paper.

“Authors described that the incorporation of pectin into vaterite microspheres enabled to reach antioxidant effect when neutrophils were activated by mucin-treated microparticles, presumably via exposed ligands. Reviewer thinks that the data demonstrating the in vivo benefit of pectin into vaterite microspheres is weak”. 

It is difficult to predict in vivo benefits of vaterite microparticles modified with pectin (CCP) on the basis of in vitro experiments. There are some points which look encouraging. Thus, with regard to development of new drug delivery systems the hybrid pectin-vaterite microparticles demonstrate a number of valuable properties. Vaterite and pectin are natural materials, both biocompatible and biodegradable [Trofimov et al 2020; Trushina et al 2022; Hadji H et al 2022; Paliya et al 2023].  Moreover, they are pH-sensitive due to vaterite matrix [Balabushevich 2022]; thus, pH-sensitivity of the drug delivery system could play an important role in the colon of patients with inflammatory bowel disease (IBD) where pH is lower than in normal colon (up to 2.3 vs 6.8) [Fallingborg 1993].  

We have shown that in line with vaterite microparticle [Balabushevich et al 2019], hybrid microparticles with pectin easily bind mucin (Table 1 and Fig. 5) which could provide their affinity to mucosa. Compared with vaterite microparticles, CCP are negatively charged that could increase their adherence to positively charged proteins in inflamed tissue [Nagashima R 1981].

Also, the diameter of CCP microparticle was less than diameter of vaterite microparticles CC (Table 1) which could facilitate their local effects in mucosa [Lamprecht et al 2001].

(This material was added to Discussion and References section).

Taken together, our results look promising for further in vivo experiments which are currently planned.

“In comparing CCP and CC, there was no difference in cytotoxicity, and while neutrophil activity decreased with CCP, SG-positive neutrophils, an indicator of Netosis, increased with CCP. There is a lack of consideration regarding the relationship between Netosis and neutrophil activity. CCP appears to be toxic in vivo, making it difficult to find consistency in the results”.

In the absence of mucin in the measurement buffer, there was no difference between CC and CCP in their activating ability towards neutrophils (Fig. 6 b). With mucin, CC activated neutrophils to higher extent than CCP. We consider this property of CCP very promising since it could prevent excessive ROS production by neutrophils in inflamed colon tissue [Simmonds et al 1992], for example. At the same time, CC and CCP particles did not cause neutrophil disfunction, even after mucin adsorption (Fig. 6 d, e). Increase in percentage of SG-positive neutrophils (Fig. 8a, c) in line with data of LDH test (Fig. 6 f) and on neutrophil PMA-induced response (Fig. 6 d, e) could be attributed to vital NETosis, this is, to early nonlytic NET formation in response to pathogen associated molecular patterns, without neutrophil death. The role of NETs in inflammation, for example, in IBD can be beneficial or negative [Drury B et al 2021] and further research in vivo is needed.  

(This material was added to Discussion section).

“It was understood that free pectin has an antioxidant effect, and it was also understood that Table 2 shows that free pectin and CCP-M have a synergistic effect on neutrophil activity. Therefore, regarding Netosis, the synergistic effect of CCP-M and free pectin should be verified as well”.

Soluble pectin significantly inhibited PMA-induced NETs formation, but did not influence spontaneous NETs formation (fig. 8 a, b). Unlike PMA, CC- and CCP-induced NETs formation was not inhibited by soluble pectin (1 mg/mL), even after mucin adsorption: in the presence of pectin F504/523 increased to 133±23% for CCP but did not change for CCP-M (102±16%). For CC and CC-M the effects were more significant: 250±79% and 219±35%. We suppose that pectin adsorption onto the particles but not pectin antioxidant properties underlie these effects. Indeed, it is another proof of non-lytic mechanism NETs formation, caused by pectin exposed on the particles, and this mechanism is known to be independent of ROS formation [Kenny et al 2017].

(This material was added to Results and Discussion section).

“The + and - symbols in Figure 8 are misaligned”.

Thank you, it was corrected

Reviewer 2 Report

This is a well-written manuscript that demonstrates the effect of added pectin on adsorbed mucin. 

Author Response

To Reviewer 2.

We are grateful to reviewer for the appreciation of our work and for the interest to the subject.

Reviewer 3 Report

The work is quite neat in presentation and experiment, but only moderately interesting.

The Discussion is very short and dry and only related to technicalities, and should be improved.

What was the motivation for the study?

What migt be the applications and consequence of the study?

Any clinical relevance?

Any potential in human disease?

English is fine.

Author Response

Response to Reviewer 3.

We are grateful to the Reviewer for constructive comments and valuable suggestions, which help improve the quality of the paper.

“The work is quite neat in presentation and experiment, but only moderately interesting.

The Discussion is very short and dry and only related to technicalities, and should be improved.

What was the motivation for the study?

What might be the applications and consequence of the study?

Any clinical relevance?

Any potential in human disease?”

We have extended Discussion with the following material:

Our study was aimed at evaluation of biocompatibility of pectin-vaterite hybrid microparticles as mucosal drug delivery system. Thus, they can be used for intranasal [16] and peroral [17] delivery, in ophthalmology [18] and pulmonology [19]. It is difficult to predict in vivo benefits of vaterite microparticles modified with pectin (CCP) on the basis of in vitro experiments, but there are some points which look encouraging. Thus, with regard to development of new drug delivery systems the hybrid pectin-vaterite microparticles demonstrate a number of valuable properties. Vaterite and pectin are natural materials, both biocompatible and biodegradable [Trofimov et al 2020; Trushina et al 2022; Hadji H et al 2022; Paliya et al 2023].  Moreover, they are pH-sensitive due to vaterite matrix [Balabushevich 2022]; thus, pH-sensitivity of the drug delivery system could play an important role in the colon of patients with inflammatory bowel disease (IBD) where pH is lower than in normal colon (up to 2.3 vs 6.8) [Fallingborg 1993]. 

We have shown that in line with vaterite microparticle [Balabushevich et al 2019], hybrid microparticles with pectin easily bind mucin (Table 1 and Fig. 5) which could provide their affinity to mucosa. Compared with vaterite microparticles, CCP are negatively charged that could increase their adherence to positively charged proteins in inflamed tissue [Nagashima R 1981].

Also, the diameter of CCP microparticle was less than diameter of vaterite microparticles CC (Table 1) which could facilitate their local effects in mucosa [Lamprecht et al 2001].

If we consider CC or CCP as potential systems for mucosal delivery, their interaction with mucin is a very important factor.

In the absence of mucin in the measurement buffer, there was no difference between CC and CCP in their activating ability towards neutrophils (Fig. 6 b). With mucin, CC activated neutrophils to higher extent than CCP. We consider this property of CCP very promising since it could prevent excessive ROS production by neutrophils in inflamed colon tissue [Simmonds et al 1992], for example. At the same time, CC and CCP particles did not cause neutrophil disfunction, even after mucin adsorption (Fig. 6 d, e). Increase in percentage of SG-positive neutrophils (Fig. 8a, c) in line with data of LDH test (Fig. 6 f) and on neutrophil PMA-induced response (Fig. 6 d, e) could be attributed to vital NETosis, this is, to early nonlytic NET formation in response to pathogen associated molecular patterns, without neutrophil death. The role of NETs in inflammation, for example, in IBD can be beneficial or negative [Drury B et al 2021] and further research in vivo is needed. 

Round 2

Reviewer 1 Report

I understood the authors' opinion. Reviewer looks forward to further studies on the biocompatibility and biological effects of CCP. I suggest acceptance of the presented revised manuscript.